# Preparation of Microcrystalline Cellulose/N-(2-aminoethyl)-3- Aminopropyl Methyl Dimethoxysilane Composite Aerogel and Adsorption Properties for Formaldehyde

**DOI:** 10.3390/polym15153155

**Published:** 2023-07-25

**Authors:** Yaning Li, Zhongzheng Liu, Chuanxi Chi, Bin Yuan, Yang Zhang, Guiquan Jiang, Jianxi Song

**Affiliations:** 1Key Laboratory of Wooden Materials Science and Engineering of Jilin Province, Beihua University, Jilin City 132013, China; liyaning@beihua.edu.cn (Y.L.); liuzhongzheng@beihua.edu.cn (Z.L.); chichuanxi@beihua.edu.cn (C.C.); yuanbin@beihua.edu.cn (B.Y.); zhangyang@beihua.edu.cn (Y.Z.); 2Key Laboratory for the Structure and Function of Polysaccharides in Traditional Chinese Medicine, Beihua University, Jilin City 132013, China

**Keywords:** cellulose, aerogel, adsorption, formaldehyde

## Abstract

Air pollution is related to the development of the national economy and people’s livelihoods. Formaldehyde, as one of the main pollutants in the air, affects people’s physical and mental health. In order to remove formaldehyde and better protect the health of residents, it is necessary to develop efficient adsorption materials. In this study, APMDS-modified cellulose composite aerogel microcrystalline was investigated. The adsorption of formaldehyde by the MCC/APMDS (Microcrystalline Cellulose/N-(2-aminoethyl)-3- Aminopropyl Methyl Dimethoxysilane) composite aerogel mainly relied upon the reaction of the protonated –NH_3_^+^ group in APMDS with formaldehyde to form a Schiff base to achieve the effect of deformaldehyde. Meanwhile, the modification of the aerogel reduced the pore volume and specific surface area, and the average pore size increased to 14.56 nm, which enhanced the adsorption capacity of formaldehyde, and the adsorption amount reached 9.52 mg/g. This study provides valuable information for the preparation of adsorbent materials with high formaldehyde adsorption capacity for air purification.

## 1. Introduction

According to research, per capita indoor stay time usually exceeds 90% [1,2,3,4]. Formaldehyde is the poster child for indoor pollutants, and more than 65% of formaldehyde is used in the production of synthetic resins in building materials [5]. There is significant evidence that formaldehyde has a positive and important effect on the development of nasopharyngeal cancer [6,7,8]. Indoor formaldehyde pollution sources are widespread. Urea-formaldehyde resins, melamine resins, phenolic resins used as adhesives for furniture boards [9,10,11], paint, lampblack and smoking in home kitchens [12] are the most common sources [13]. In addition, formaldehyde will make indoor residents experience headaches, nausea, and mucosal inflammation; nasopharynx, eye and throat irritation; asthma and allergic rhinitis; eczema, fatigue, etc. [13,14]. Long-term inhalation of formaldehyde can also lead to lower learning and work efficiency and have an unfavorable effect on the physical and mental health of residents [15,16,17].

So far, several indoor formaldehyde removal strategies have been developed, such as adsorption, plasma-catalyzed decomposition, photo-catalytic degradation and plant purification methods. Adsorption is one of the most convenient methods for indoor formaldehyde removal. Its advantages mainly lie in its fast removal rate, high efficiency, simple operation, energy savings and low cost. Usually, adsorption methods are divided into physical adsorption and chemical adsorption methods. Physical adsorption relies on weak intermolecular forces between the adsorbent and formaldehyde, and therefore, its stability is poor. In contrast, chemical adsorption relies on the surfactant functional group of the adsorbent forming a strong chemical bond with HCHO. As a result, it is highly stable and selective. Therefore, many adsorbents, such as SiO_2_ [18,19], Al_2_O_3_ [20], MOFs [21] and microporous carbonaceous materials [22,23], have been widely studied for indoor formaldehyde adsorption and removal. In recent years, various novel materials, such as active carbon fibers (ACFs), carbon nanotubes (CNTS), graphene, metal–organic skeletons (MOF) and porous organic polymers (pops) [7,9,22,24,25], have been introduced as effective adsorbents for removing HCHO. The adsorption performance of nitrogen-containing porous materials for HCHO is better than that of common adsorbents, and the adsorption performance is closely related to the type and content of surface nitrogen [26]. Therefore, surface modification of adsorbents is an effective way to improve adsorption capacity by increasing the adsorption’s active site and the selectivity of HCHO.

Amino-modified cellulose aerogel adsorbents show a high adsorption capacity of HCHO. APMDS is often used for the functionalization of cellulose nanofibers to enhance the adsorption capacity of cellulose aerogel. APMDS-modified cellulose aerogels are usually achieved through two different methods. One is the synthesis of amino-silanized cellulose by freeze-drying cellulose and APMDS suspension, resulting in the the preparation of amino-silanized cellulose aerogel. Li et al. [27] reacted CNF suspension with APMDS suspension to prepare APMDS-modified CNF and finally prepared modified CNF aerogel through ultrasonic treatment and freeze-drying. Another approach is to functionalize cellulose hydrogels or cellulose membranes in APMDS solution and then proceed with supercritical carbon dioxide drying or freeze-drying to prepare. Zhang et al. [28] reacted prepared cellulose hydrogel spheres in 12 wt% APMDS solution for 12 h (100 °C) and finally obtained amino-modified A-NCC aerogel by freeze-drying.

Compared with traditional adsorption materials, nano-cellulose-based aerogels have better adsorption capacity, and their adsorption performance for formaldehyde can be further improved by means of amino functionalization. As an adsorbent, nano-cellulose-based aerogels are a potential field for current and future research.

## 2. Preparation of MCC/APMDS Composite Aerogel

Dissolving 0.1 g microcrystalline cellulose in 5 g 60 wt% LiBr solution (1:100), we magnetically stirred the solution until it was evenly dispersed. We then heated it in a 140 °C oil bath on high for 20 min until a clear homogeneous solution was formed. We poured the hot cellulose suspension into the mold and then used an ultrasonic cleaner to remove air bubbles from the cellulose suspension. After that, we cooled to room temperature to form a gel. We soaked the formed gel in deionized water and rinsed it with excessive deionized water until lithium bromide could not be detected by silver nitrate solution. The gel was then solvent displaced using ethanol and tert-butanol in an attempt to replace the water in the hydrogel with tert-butanol. Finally, it was frozen in an ultra-low-temperature freezer at −80 °C for 12 h. After removing the frozen sample, we placed it in a freeze dryer to obtain cellulose aerogel.

The cellulose hydrogels and tert-butanol (mass ratio 1:4) prepared above were placed in a 250 mL round-bottomed flask, and the pH of the mixture was adjusted to 4–5 by adding acetic acid to facilitate the hydrolysis of APMDS. Then, a certain amount of APMDS (0, 2, 4, 6 and 8 wt%, based on the total mass of the cellulose hydrogel and tert-butanol mixture) was added to this mixture, and the mixture was reacted at 90 °C for 4 h. After the reaction was completed, the APMDS and its self-condensation were removed by repeated washing three to five times with the tert-butanol solution. Meanwhile, solvent replacement was carried out, and the gel after solvent replacement was freeze-dried for 24 h to obtain MCC/APMDS composite aerogel. The above composite aerogels modified by APMDS with different mass fractions (2, 4, 6 and 8 wt%) were denoted as ACC-1, ACC-2, ACC-3 and ACC-4, respectively. The aerogel prepared with a mass fraction of 0 wt% was denoting as CAB aerogel.

According to the above method, 6 wt% APMDS was added to the mixture solution of cellulose hydrogel and tert-butanol (mass ratio 1:4). The reaction temperature was controlled at 90 °C, and the reaction time was set as 1, 2, 3, 4 and 5 h. After the reaction was completed, the MCC/APMDS composite aerogel with different reaction times (1, 2, 3, 4 and 5 h) was washed, solvent replacement was carried out, and the gel after solvent replacement was freeze-dried in accordance with the above methods to obtain ACC-1 h, ACC-2 h, ACC-3 h, ACC-4 h and ACC-5 h, respectively.

According to the above method, 6 wt% APMDS was also added to the mixture solution of cellulose hydrogel and tert-butanol (mass ratio 1:4). The reaction time was controlled as 4 h, and the reaction temperature was set as 60, 70, 80, 90 and 100 °C. After the reaction was complete, the MCC/APMDS composite aerogel with different reaction temperatures (60, 70, 80, 90 and 100 °C) was washed, solvent replacement was carried out, and the gel after solvent replacement was freeze-dried according to the above methods. Finally, the composite aerogel samples were obtained as ACC-60, ACC-70, ACC-80, ACC-90 and ACC-100, respectively.

## 3. Characterization Details

The morphology of the sample was observed by scanning electron microscope (JSM 7600F, Ltd., Tokyo, Japan), where the current was 10 μA, and the acceleration voltage was 5 kV. The structure of the sample was analyzed by an infrared spectrometer, where the scanning range was 4000–400 cm^−1^, and the resolution was 4 cm^−1^. After drying the aerogel sample at 60 °C for 12 h, the pore structure of the sample was determined by a specific surface area and porosity meter (ASAP 2020, Micromeritics Instrument Corporation, Norcross, Norcross, GA, USA). An X-ray diffractometer (D/max-2500VL/PC, Rigaku Corporation, Tokyo, Japan) was used to characterize the crystalline state of the corresponding aerogel samples and cellulose raw materials, where the acceleration voltage was 40 KV, the current was 50 mA, the scanning speed was 4°/min, and the scanning range was 5°–40°. The thermal stability of the corresponding aerogel samples and cellulose samples was analyzed by a thermos gravimetric analyzer (NETZSCH STA409PC, Netzsch Gerätebau GmbH, Selb, Germany), where the heating rate was 10 K/min, and the measuring range was 40–600 °C. X-ray photoelectron spectroscopy (THERMO, Thermo Fisher Scientific, Wilmington, NC, USA) was used to analyze the surface of the cellulose aerogel samples before and after modification.

## 4. Analysis of Adsorption of Gaseous Formaldehyde

A static adsorption experiment of gaseous formaldehyde was carried out on the prepared sample at room temperature. The 0.05 g aerogel sample was weighed and then dried in a vacuum drying oven at 80 °C for 12 h. We then put it into the transition chamber of a glove box (170 L). Weighing out 20 mg of paraformaldehyde, we put this into a round-bottomed flask, heat it at 70 °C in a water bath for 1 h and passed the formaldehyde gas generated by the reaction into the reaction chamber of the box. After the concentration of gaseous formaldehyde in the reactor became stable for 1 h, we opened the transition chamber and placed the weighed aerogel sample into the reaction chamber for adsorption. We placed the portable formaldehyde detector (PPM HTV in the UK) into the reaction chamber of the glove box in advance. After the aerogel sample began to adsorb, the sampling interval was 30 min in the early stage and 60 min in the late stage of adsorption. We then determined the concentration of formaldehyde and calculated the adsorption amount. The adsorption amount of formaldehyde was calculated using the following formula:(1)$$q_{t}=\frac{{(C}_{0}-C_{t})V}{m}$$

In the formula, $q_{t}$ is the adsorption amount of formaldehyde at t time, mg/g; $C_{0}$ and $C_{t}$ are the initial concentration and t time concentration of formaldehyde, mg/m^3^; m is the mass of the aerogel, mg; V is the volume of the glove box, m^3^.

## 5. Results and Discussion

### 5.1. SEM Analysis

Figure 1 shows a comparison of the microscopic morphologies observed by SEM of the CAB aerogel (Figure 1a) and the ACC-3 (Figure 1b) composite aerogel. It can be seen that the addition of APMDS did not significantly change the shape of the internal structure of the cellulose aerogel. The aerogel samples all showed a three-dimensional network structure with irregularly shaped nano-scale pores. The aerogel observed in Figure 1 is highly porous, indicating that the pore morphology did not collapse during processing. The CAB aerogel had a low density (21.8 mg/cm^3^) and high porosity (98.2%) compared with the uniform porosity of the CAB aerogel, as with the introduction of APMDS, a large number of pores in the aerogel are blocked and cross linked, forming a flaky structure. This phenomenon may be due to the introduction of APMDS, which promotes the aggregation of cellulose during freezing and requires a certain space for the cross linking and aggregation of amine-loaded groups, thereby causing the pores to be blocked. From the comparison, it was found that Si and N elements appeared on ACC-3, which indicated that ADMPS had been successfully grafted onto the aerogel.

### 5.2. N_2_ Sorption Isotherms and BET Analysis, FT-IR Analysis, XRD Analysis

As shown in Figure 2a, similar isotherm curves were obtained, and the aerogel before and after modification had H1-type retention rings of type IV isotherms, indicating the typical mesoporous structure of the aerogel. P/P0 formed a large retention ring structure in the range of 0.8 to 1.0, which revealed the existence of the mesoporous structure in the aerogels, with abundant mesoporous and microporous. In addition, the pore size distribution curve shown in Figure 2b shows that the pore size of the aerogel before and after modification was concentrated in the range of 2–20 nm, and the peak value of the MCC/APMDS composite aerogel moved to the left compared to the CAB aerogel. This is possibly due to the change in pore structure distribution resulting from the grafting of APMDS to the cellulose chain. After APMDS was grafted onto the cellulose, the specific surface area and pore distribution of the aerogel were changed. After APMDS modification, the pore volume and specific surface area of the aerogel were reduced. By using BJH and BET methods to calculate the characteristics of pore parameters, it was concluded that the specific surface area and pore volume of the CAB aerogel and the ACC-3 aerogel were 127.1 m^2^/g and 0.272 cm^3^/g and 102.8 m^2^/g and 0.212 cm^3^/g, respectively. The average pore size of the modified aerogel increased from 12 nm to 14.56 nm, meaning that some may be covered. The results are consistent with those obtained from the examination of SEM images.

As shown in Table 1, the aerogel before and after modification is a lightweight material with high porosity. However, the density of the aerogel after amination modification with APMDS increased, and its porosity decreased. This may be because the introduction of APMDS leads to an increase in the mass of the cellulose aerogel, the blockage of pores inside the aerogel and a decrease in spatial structure, which leads to an increase in the density and a decrease in the porosity of the modified aerogel.

Figure 2c shows the FTIR spectra of the CAB aerogel and the aerogel modified with APMDS. As shown, the infrared absorption spectra of both the modified and unmodified aerogels show bands typical of cellulose. For example, the broad absorption peak at 3430 cm^−1^ is the stretching vibration peak of hydroxyl groups between cellulose molecules. The absorption peak near 2926 cm^−1^ is the stretching vibration peak of C–H, and the bending vibration peak of –CH_2_ is the absorption peak near 1429 cm^−1^. The absorption peak near 1163 cm^−1^ is the asymmetric stretching vibration peak of C–O–C ether in the pyran ring. The absorption peak around 1060 cm^−1^ can be attributed to the absorption vibration peak of C–O. These results show that the aerogel modified by APMDS still has the matrix materials of cellulose.

In addition, some important changes in the infrared spectrum were observed after modification with APMDS compared to the IR spectra of the CAB aerogel. There is a new peak at 1261 cm^−1^, which is usually related to C–O–Si tensile vibration. Due to the introduction of the N–H bond after modification, with the increase in APMDS concentration, the band intensity around 2926 cm^−1^ increases. In addition, two new bands were observed at 795 and 1575 cm^−1^, which were attributed to N–H bending vibration and N-H stretching vibration of –NH_2_ after amination. All these results showed that APMDS was successfully grafted onto CNF, and the primary amine group of APMDS remained intact.

Figure 2d shows the XRD curves of the CAB and ACC-3 aerogel samples. In Figure 2d, diffraction peaks at 2θ = 12.1° and 20.2° can be seen, and these diffraction peaks correspond to the of type Ⅱ cellulose, respectively. The characteristic curves of the ACC-3 aerogel show the characteristic peaks of cellulose. This indicates that the crystal structure of cellulose was not changed by the introduction of APMDS.

### 5.3. XPS Analysis, TG Analysis

Figure 3 shows the XPS spectra using the CAB and MCC/APMDS composite aerogels, respectively. It can be seen from the broad sweep spectra that an amino silane grafting reaction occurred in the MCC/APMDS composite aerogel after modification with APMDS, and the Si element appeared in the broad sweep spectra, which is consistent with the results of infrared analysis. The low-resolution spectra of the unmodified CAB aerogel showed that C and O atoms were the main components in the aerogel, whereas N and Si silicon elements were detected in the modified aerogel, indicating successful grafting of APMDS with cellulose. As can be seen from the high-resolution maps of C1s and N1s of ACC-3 (illustration in Figure 3b), there are four types of carbon bonds: C–C (C1, 284.8 eV), C–O (C2, 286.4 eV), O–C–O (C3, 287.8 eV) and C–N (C4, 288.9 eV), respectively. The presence of the N element is divided into two forms, namely, –NH_2_ and –NH_3_^+^. This indicates that the primary amino group was protonated during chemical grafting. The energy peak position of O in Figure 3c is 531 eV, the peak shape of O is a slightly asymmetric peak, and a low-energy shoulder slit can be observed, suggesting that oxygen is present in the form of hydroxide ions (OH–), which is speculated to be the result of Schiff base formation.

As shown in Figure 3d, the MCC/APMDS composite aerogel has two thermal degradation processes. The first thermal degradation process is between 50 °C and 180 °C. In this temperature range, cellulose aerogel has slight thermal degradation, and the absorbed water inside the aerogel overflows, resulting in slight mass loss. Thermal degradation was observed between 200 °C and 400 °C, which may be related to the breaking of cellulose glycosides bonds; the inside of the aerogel collapsed and degraded, which produced a lot of water and gas, and then the cellulose aerogel lost a lot of weight. The residual weight loss amount of the aerogel modified with APMDS is larger than that of the CAB aerogel. At the same time, with the increase in the use of APMDS, the greater the content of nitrogen, and the greater the residual weight loss amount. This may be due to the possible existence of the Si element in APMDS; the modified CNC aerogel generates refractory substances, such as SiO_2_, during thermal degradation, thus increasing the thermal degradation residues of the aerogel. Because the higher the content of nitrogen, the more refractory substances produces, so the weight loss residue of the MCC/APMDS composite aerogel increases, and the maximum residue reaches 25.6%. The results also indirectly indicated that AEAPMDS was successfully grafted onto cellulose. In addition, the thermos gravimetric curve of the modified aerogel is much smoother than that of the unmodified aerogel, probably because the introduction of APMDS causes the chemical bond energy of the CAB aerogel to increase, which requires higher thermal degradation energy, and the degradation process is more moderate.

### 5.4. Effect of Different Reaction Conditions on Nitrogen Content of MCC/APMDS Composite Aerogel Samples

A single-factor method was used to investigate the effects of the APMDS dosage (based on the total mass of the mixture solution), the reaction time and the reaction temperature on the nitrogen content of the MCC/APMDS composite aerogel. At the same time, the nitrogen content of MCC/APMDS composite aerogel samples was detected by X-ray photoelectron spectroscopy, and the final results were obtained as shown in Figure 4a.

Figure 4a shows the effect of APMDS usage on the nitrogen content of the MCC/APMDS composite aerogel when the reaction time is 4 h and the reaction temperature is 90 °C. It can be seen from Figure 4a that with the increase in the amount of APMDS added, the nitrogen content of the sample increases rapidly. However, when the amount of APMDS was greater than 6%, there was almost no increase in nitrogen content. This may be because the higher amount of APMDS may lead to the rapid formation of APMDS self-condensation polymer, which is not conducive to the reaction of APMDS with cellulose. Therefore, nitrogen content increased slowly after the application of 6 wt% APMDS. The results showed that the appropriate amount of APMDS was 6 wt%, and the nitrogen content of the MCC/APMDS composite aerogel reached 8.98 wt% when 6 wt% APMDS was used.

Figure 4b shows the effect of reaction time on the nitrogen content of the MCC/APMDS composite aerogel when the amount of APMDS is 6 wt% and the reaction temperature is 90 °C. As shown in Figure 4b, the nitrogen content of the MCC/APMDS composite aerogel increased rapidly with the increase in reaction time before 3 h. But at 4 h, almost all reached their own plateau. This may be because before the reaction time of 3 h, due to the reaction of a large number of reactive hydroxyl groups on cellulose and the high content of free amino silanols formed by the hydrolysis of APMDS in the reaction medium, the content of nitrogen elements increases rapidly. After the reaction time reaches 4 h, the reactive hydroxyl groups or APMDS reactions on cellulose are exhausted. As a result, the nitrogen content of the MCC/APMDS composite aerogel remained constant after 4 h of reaction. The results showed that the nitrogen content of the MCC/APMDS composite aerogel reached the maximum when the reaction time was 4 h.

Figure 4c shows the effect of the reaction temperature on the nitrogen content of the MCC/APMDS composite aerogel when the amount of APMDS is 6 wt% and the reaction time is 4 h. When the reaction temperature is lower than 70 °C, the nitrogen content of the MCC/APMDS composite aerogel increases very slowly with the increase in temperature. However, After 70 °C there is a significant increase until the reaction temperature rises to 90 °C. After 90 °C, little effect of the reaction temperature on the amine load was observed. This is because APMDS is prone to hydrolysis to form amino silanols even at room temperature, so the results suggest that temperatures higher than 70 °C are needed to facilitate the reaction of amino silanols with hydroxyl groups on cellulose. The rapid increase in the nitrogen content of the MCC/APMDS composite aerogel after the reaction temperature was 70 °C may be due to the increase in temperature, which increases the rate constant of the reaction of APMDS with the hydroxyl group on the cellulose. At temperatures greater than 90 °C, the nitrogen content of the MCC/APMDS composite aerogel has little effect. From Figure 4d, it can be seen that as the equilibrium concentration increases, the amount of adsorption is greater and greater. It remains stable after reaching a certain concentration, and the highest amount of adsorption is found when the temperature is 90 °C. These results indicate that the optimum temperature is 90 °C.

From the above results, it can be seen that the nitrogen content of the MCC/APMDS composite aerogel reaches its maximum when the amount of APMDS is 6 wt%, the reaction time is 4 h, and the reaction temperature is 90 °C.

### 5.5. Formaldehyde Adsorption Performance Analysis

The adsorption results of formaldehyde gas by the MCC/APMDS composite aerogel are shown in Figure 5. The experiment was conducted at 25 °C, and the initial concentration of formaldehyde was 5.0 mg/m^3^. Figure 5a–c show the results of the adsorption of formaldehyde gas by the MCC/APMDS composite aerogel by the amount of APMDS, reaction time and reaction temperature, respectively. It can be seen from the Figure 4 that after formaldehyde adsorption by the MCC/APMDS composite aerogel for 2 h, the sample tends to be saturated, and its maximum adsorption capacity is 9.52 mg/g. As can be seen from Figure 5a, the formaldehyde adsorption capacity of the modified aerogel increased with the increase in the concentration of the modifier. When the amount of APMDS reached 8 wt%, the adsorption capacity of the aerogel for formaldehyde gas hardly increased compared with that of the 6 wt%. As can be seen from Figure 5b, with the increase in reaction time, the adsorption amount of formaldehyde by the aerogel increases, and when the reaction time reaches 5 h, the formaldehyde adsorption amount hardly increases. It can be seen from Figure 5c that the formaldehyde adsorption capacity increases with the increase in temperature. When the reaction temperature is lower than 80 °C, the formaldehyde adsorption capacity of the aerogel increases slowly, and when the temperature reaches 100 °C, the formaldehyde adsorption capacity hardly increases. This phenomenon is related to the change in nitrogen content in the MCC/APMDS composite aerogel.

Table 2 shows that the adsorption amount is related to the nitrogen content, and the higher the nitrogen content, the stronger the adsorption in a certain range. As can be seen from Table 3 and Table 4, the sample prepared under the conditions of an APMDS dosage of 6 wt%, a reaction time of 4 h and a reaction temperature of 90 °C had the largest average pore size (14.56 nm) and the largest formaldehyde adsorption capacity (9.52 mg/g). After many repeated tests, this sample was found to be able to keep the adsorption amount and the average pore size error very small and to remain stable.

As can be seen from Figure 6a,b, after the adsorption of formaldehyde by the MCC/APMDS composite aerogel, part of the microporous pores become smaller and covered up, which proves that the aerogel modified by APMDS has good adsorption performance on formaldehyde. The black line in Figure 7a and the purple line in Figure 7b both represent element C. The different N and H distributions lead to the appearance of two different morphologies, –NH_3_^+^ and –NH_2_. According to Figure 7a,b, when the MCC/APMDS composite aerogel adsorbed formaldehyde, the content of –NH_3_^+^ is obviously reduced, which indicates that the positively charged nitrogen groups are closer to formaldehyde molecules. This indicates that the adsorption effect of the MCC/APMDS composite aerogel on formaldehyde mainly depends on the protonated –NH_3_^+^ group of APMDS to react with formaldehyde to produce Schiff bases to achieve the effect of formaldehyde removal.

## 6. Conclusions and Outlook

In this paper, MCC/APMDS composite aerogels were successfully prepared by heating, solvent replacement and freeze-drying using 2 wt% cellulose hydrogel as a matrix material and modified with aminosilane (APMDS). The modified MCC/APMDS composite aerogels underwent an aminosilane grafting reaction, and the adsorption effect of formaldehyde was mainly dependent upon the protonated –NH_3_^+^ group of APMDS reacting with formaldehyde to produce Schiff bases to achieve the effect of formaldehyde removal. Meanwhile, the modification of the aerogel resulted in a reduction in pore volume and specific surface area. An increase in the average pore size to 14.56 nm also led to a stronger adsorption capacity for formaldehyde, and the adsorption amount could reach 9.52 mg/g. This study provides valuable information for the preparation of adsorbent materials with a high efficiency of formaldehyde adsorption capacity for air purification.

## Figures and Tables

**Figure 1 polymers-15-03155-f001:**
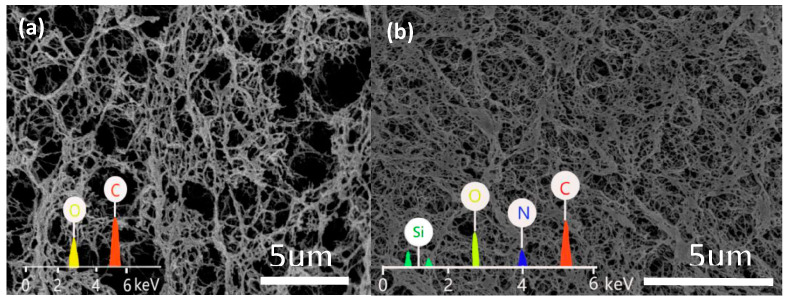
SEM-EDS diagram of CAB (**a**) and MCC/APMDS (**b**) composite aerogel.

**Figure 2 polymers-15-03155-f002:**
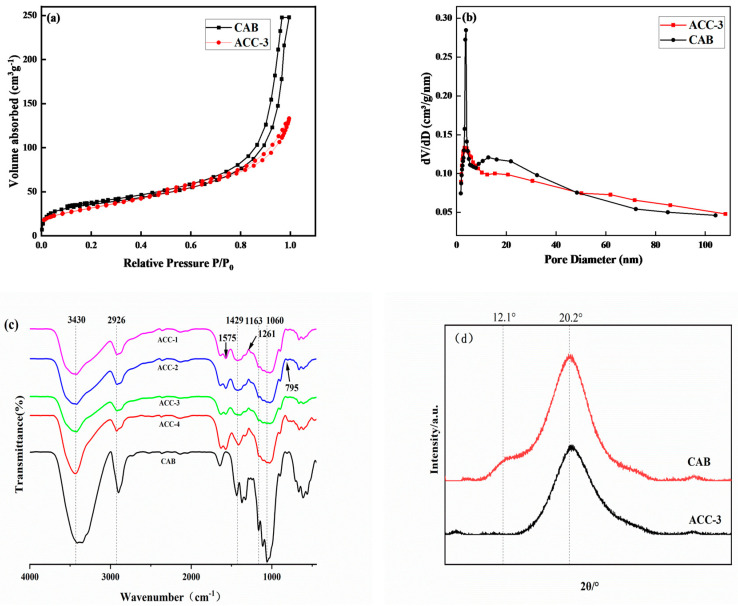
N_2_ adsorption−desorption isotherm (**a**) and pore size distribution (**b**) of aerogels. FT-IR diagram of CAB and MCC/APMDS composite aerogel (**c**); XRD diagram of CAB and MCC/APMDS composite aerogel (**d**).

**Figure 3 polymers-15-03155-f003:**
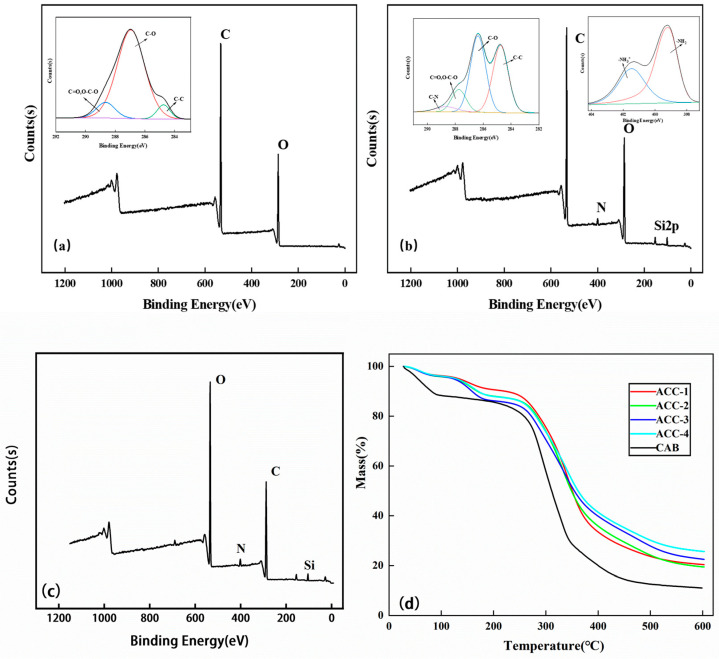
XPS spectra: (**a**) CAB pre-adsorption; (**b**) ACC-3 pre-adsorption; (**c**) ACC-3 post-adsorption; TG spectra of CAB and MCC/APMDS composite aerogel (**d**).

**Figure 4 polymers-15-03155-f004:**
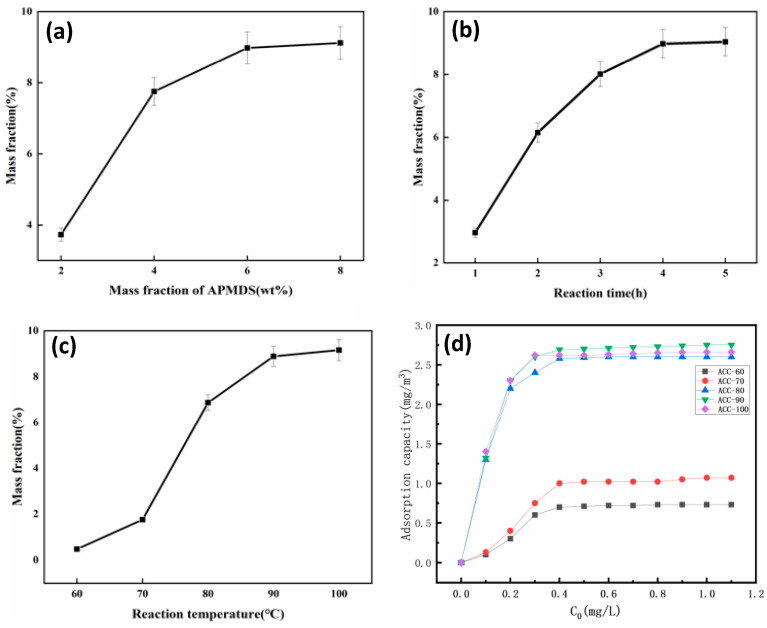
Effect of APMDS usage on nitrogen content of MCC/APMDS composite aerogel. (**a**) Effect of reaction time on nitrogen content of MCC/APMDS composite aerogel. (**b**) Effect of reaction temperature on nitrogen content of MCC/APMDS composite aerogel. (**c**) Adsorption versus equilibrium concentration curves at different temperatures (**d**).

**Figure 5 polymers-15-03155-f005:**
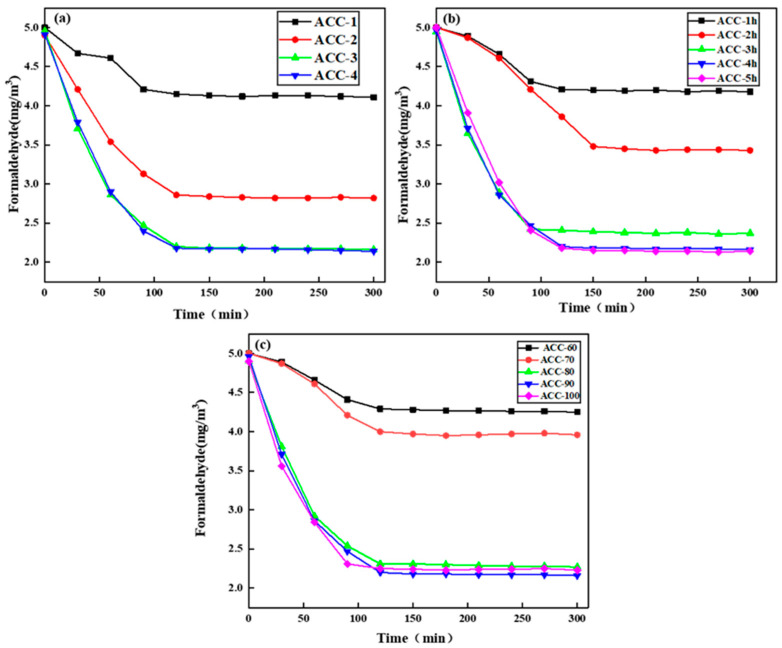
Change in formaldehyde concentration.

**Figure 6 polymers-15-03155-f006:**
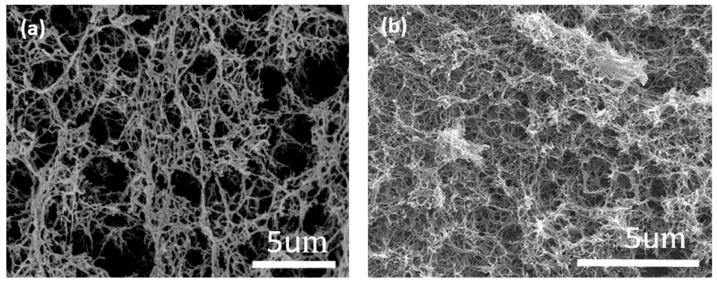
SEM diagram of sample ACC-3 before (**a**) and after (**b**) adsorption.

**Figure 7 polymers-15-03155-f007:**
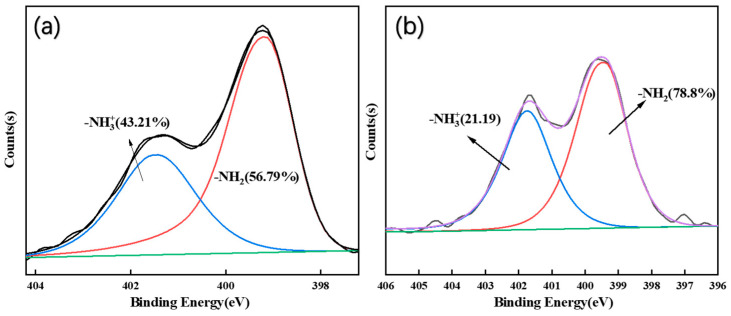
N1s high-resolution pattern of samples before and after ACC-3 adsorption: (**a**) before adsorption; (**b**) post-adsorption.

**Table 1 polymers-15-03155-t001:** Physical properties of CAB and MCC/APMDS composite aerogel.

Sample	BET Surface Area /(m^2^ g^−1^)	Average Aperture/nm	Total Pore Volume/(cm^3^·g^−1^)	Density (mg/cm^3^)
CAB	127.1	12	0.272	21.8
ACC-3	102.8	14.56	0.212	26.7

**Table 2 polymers-15-03155-t002:** Analysis of experimental errors for different variables at 95 per cent confidence level.

Variant	Tolerance Range (Nitrogen Content%)				
APMDS usage (2, 4, 6, 8%)	3.73 ± 0.1865	7.75 ± 0.3875	8.98 ± 0.449	9.12 ± 0.456	
Response time (1 h, 2 h, 3 h, 4 h, 5 h)	2.97 ± 0.1485	6.15 ± 0.3075	8.01 ± 0.4005	8.98 ± 0.449	9.04 ± 0.452
Reaction temperature (60 °C, 70 °C, 80 °C, 90 °C, 100 °C)	0.48 ± 0.024	1.76 ± 0.088	6.87 ± 0.3435	8.89 ± 0.4445	8.89 ± 0.458

**Table 3 polymers-15-03155-t003:** Mean pore size and adsorption properties of aerogels of different variables tested.

MCC/APMDS Aerogel	Average Pore Size (nm)	Adsorption Capacity (mg/g)
2%APMDS, 4 h, 90 °C	14.21	9.33
4%APMDS, 4 h, 90 °C	14.43	9.36
6%APMDS, 4 h, 90 °C	14.56	9.52
8%APMDS, 4 h, 90 °C	14.53	9.49
6%APMDS, 4 h, 60 °C	13.91	9.41
6%APMDS, 4 h, 70 °C	14.19	9.45
6%APMDS, 4 h, 80 °C	14.23	9.52
6%APMDS, 4 h, 100 °C	14.51	9.33
6%APMDS, 1 h, 90 °C	13.88	8.16
6%APMDS, 2 h, 90 °C	14.13	8.98
6%APMDS, 3 h, 90 °C	14.37	9.24
6%APMDS, 5 h, 90 °C	14.52	9.33

**Table 4 polymers-15-03155-t004:** Error analysis of repeated tests at 95 per cent confidence level.

6%APMDS, 4 h, 90 °C MCC/APMDS Aerogel	Average Pore Size (nm)	Adsorption Capacity (mg/g)
Test 1	14.56	9.49
Test 2	14.59	9.52
Test 3	14.54	9.54
Test 4	14.58	9.50
Test 5	14.53	9.55
Average value	14.56	9.52
Standard deviation	0.0228	0.0228
Tolerance range	14.56 ± 0.0102	9.52 ± 0.0102

## Data Availability

No new data were created or analyzed in this study.

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
