# Peer review of "Preparation of Microcrystalline Cellulose/N-(2-aminoethyl)-3- Aminopropyl Methyl Dimethoxysilane Composite Aerogel and Adsorption Properties for Formaldehyde"

_polymers, 2023, doi:10.3390/polym15153155_

Round 1
Reviewer 1 Report
The study presented in this paper explores the development of adsorption materials for the removal of formaldehyde. The researchers focused on a cellulose composite aerogel modified by APMDS to enhance its adsorption capacity.
The results of the study demonstrate that the modification of the aerogel led to a decrease in pore volume and specific surface area while the average pore diameter increased. These changes resulted in a stronger adsorption capacity for formaldehyde.
The researchers found that the sample prepared with an APMDS dosage of 6 wt%, a reaction time of 4 hours, and a reaction temperature of 90℃ exhibited the largest average pore size and the highest formaldehyde adsorption capacity (9.52 mg/g). Can you estimate the error associated with these measurements and round the number accordingly? It can not be in the second decimal place.
The excellent adsorption performance of the composite aerogel can be attributed to its high porosity and the presence of active amino groups on its surface. These characteristics enable the material to effectively capture and remove formaldehyde from the air. It would be nice to make a connection with XPS results before and after adsorption to construct a mechanism and present it as an additional figure.
Also, steady kinetic curves for formaldehyde adsorption allow the construction of adsorption isotherm curves, especially when the optimal temperature is determined. Please provide adsorbed quantity vs. equilibrium concentration curves.
Overall, this research is significant as it addresses the pressing issue of air pollution. The study is comprehensive, and a number of experimental techniques were used; therefore, I recommend publication after minor revision.
Minor editing of English language required
Author Response
Dear Reviewer,
Thank you for your kind letter with the constructive comments from all the
reviewers on our manuscript (entitled " Preparation of MCC/APMDS composite aerogel and adsorption properties for formaldehyde", Manuscript Number: polymers-2518863). All the comments have been thoroughly considered and corrections are marked in red in the revised manuscript. The point-to-point answers and explanations for all revisions are listed on the Response to Reviewers. Thank you for your consideration. If further information is required, please not hesitate to contact with me.
Jianxi Song Ph.D. Key Laboratory of Wooden Materials Science and Engineering of Jilin Province
Beihua University
Jilin 132013, China
Email: [email protected]

Reviewer 2 Report
In this work, the authors reported the “Preparation of MCC/APMDS composite aerogel and adsorption properties for formaldehyde”. However, this manuscript must improve in some ways before being accepted in Polymers.
1. In the title, the authors should write the full name of MCC/APMDS.
2. the schematic synthesis of MCC/APMDS composite preparation should be provided with their chemical structures.
3. The scale bar is not visible in all SEM images.
4. the chemical structures of MCC/APMDS composite aerogel should be examined by SEM-EDS mapping with their elements images.
5. the quality of all figures should be improved and some figures are so small in size.
6. Figure 2(c) is unclear.
7. Figure 3 is hard to review.
8. the important thing, the authors should provide the adsorption mechanism of MCC/APMDS composite aerogel for formaldehyde.
9. the reproducibility of MCC/APMDS composite aerogel after the HCHO adsorption process should be investigated.
10. Rewrite the conclusion part.
11. There are more typological errors, so authors need to correct them properly.
Moderate editing of English language required.
Author Response

(The authors gave the same response as above.)

Round 2
Reviewer 2 Report
This manuscript could be accepted in this journal as it is.